# A Pilot Study to Investigate the Balance between Proteases and α1-Antitrypsin in Bronchoalveolar Lavage Fluid of Lung Transplant Recipients

**DOI:** 10.3390/ht8010005

**Published:** 2019-02-13

**Authors:** Maddalena Cagnone, Davide Piloni, Ilaria Ferrarotti, Monica Di Venere, Simona Viglio, Sara Magni, Anna Bardoni, Roberta Salvini, Marco Fumagalli, Paolo Iadarola, Sabrina Martinello, Federica Meloni

**Affiliations:** 1Department of Molecular Medicine, Biochemistry Unit, University of Pavia, 27100 Pavia, Italy; maddalena.cagnone@gmail.com (M.C.); monica.divenere01@universitadipavia.it (M.D.V.); simona.viglio@unipv.it (S.V.); abardoni@unipv.it (A.B.); roberta.salvini@unipv.it (R.S.); 2Department of Internal Medicine and Therapeutics, Section of Pneumology, University of Pavia, 27100 Pavia, Italy; davidepiloni@live.it (D.P.); I.Ferrarotti@smatteo.pv.it (I.F.); sabrinamartinello@hotmail.it (S.M.); f.meloni@smatteo.pv.it (F.M.); 3IRCCS Foundation Policlinico San Matteo, Department of Medical Sciences and Infective Diseases, Unit of Respiratory Diseases, 27100 Pavia, Italy; sara.magni04@gmail.com; 4Department of Biology and Biotechnologies “L. Spallanzani″, Biochemistry Unit, University of Pavia, 27100 Pavia, Italy; marco.fumagalli@unipv.it

**Keywords:** alpha 1-antitrypsin, bronchoalveolar lavage fluid, bronchiolitis obliterans syndrome, neutrophils, elastase, lung

## Abstract

The neutrophilic component in bronchiolitis obliterans syndrome (BOS, the main form of chronic lung rejection), plays a crucial role in the pathogenesis and maintenance of the disorder. Human Neutrophil Elastase (HNE), a serine protease responsible of elastin degradation whose action is counteracted by α1-antitrypsin (AAT), a serum inhibitor specific for this protease. This work aimed to investigate the relationship between HNE and AAT in bronchoalveolar lavage fluid (BALf) from stable lung transplant recipients and BOS patients to understand whether the imbalance between proteases and inhibitors is relevant to the development of BOS. To reach this goal a multidisciplinary procedure was applied which included: (i) the use of electrophoresis/western blotting coupled with liquid chromatography-mass spectrometric analysis; (ii) the functional evaluation of the residual antiprotease activity, and (iii) a neutrophil count. The results of these experiments demonstrated, for the first time, the presence of the complex between HNE and AAT in a number of BALf samples. The lack of this complex in a few specimens analyzed was investigated in relation to a patient’s lung inflammation. The neutrophil count and the determination of HNE and AAT activities allowed us to speculate that the presence of the complex correlated with the level of lung inflammation.

## 1. Introduction

Much of the lung destruction in acute and chronic lung diseases is caused by the deleterious activities of multiple proteases, among which human neutrophil elastase (HNE) is the serine protease mostly responsible of elastin degradation [1,2]. HNE is a serine protease secreted by neutrophils and it was shown to orchestrate inflammation through the activation of cathepsin B and matrix metalloprotease 2 (MMP-2) whose upregulation contributes to pulmonary infection and tissue destruction [3,4]. Given this role of HNE, it can be postulated that its neutralization with a specific anti-protease would automatically inhibit the upregulation of other proteases, thus lessening the overall protease burden. This could prevent/decrease lung destruction and contribute to the restoration of lung host defense [3]. The specific inhibitor of HNE is α1-antitrypsin (AAT), a 52 kDa glycoprotein mainly produced and secreted (70–80%) by hepatocytes. In small amounts, it is also synthesized by monocytes, macrophages, pulmonary alveolar cells and by intestinal and corneal epithelium [5,6,7,8]. AAT provides more than 90% of the anti-proteinase activity in human serum, the remaining 10% being provided by β-2 macroglobulin. Inherited α1-antitrypsin deficiency (AATD) is a clear example that the balance between proteases and antiproteases is one of the most important aspects of lung homeostasis. Being that the increased levels of HNE positively correlated with the severity of emphysema, a careful monitoring of the ratio between the amount of protease and its specific inhibitor is vital for these patients [9,10]. 

Despite the high specificity of AAT for HNE, additional skills which support the key role of this inhibitor in regulating inflammatory response have recently been evidenced. Numerous reports, in fact, have shown its ability to neutralize a broad range of other serine proteases and different classes of proteases, which include proteinase-3, myeloperoxidase, cathepsin G (Cat. G), metalloproteases and cysteine-aspartic proteases [11,12,13,14,15,16,17]. 

The role of the AAT/HNE balance in homeostasis of the lung was investigated by Pajdak et al. [14] by means of an immune-electrophoretic procedure. This approach allowed researchers to observe the presence of AAT-HNE complexes in bronchoalveolar lavage fluid (BALf) from patients with inflammatory lung disease, asthma and lung cancer.

More recently, both HNE activity and AAT concentration have been quantified by Hirsch et al. [16] in a cohort of lung transplant recipients by means of a titration assay and an enzyme-linked immunosorbent assay (ELISA). Their study was focused on the possible role of alterations in antiprotease defense and of oxidant activities for the pathogenesis of chronic rejection after lung transplantation, an increasingly widely applied therapy for end-stage lung diseases in the past two decades.

Long-term survival of a lung graft is hampered by the occurrence of chronic lung allograft dysfunction (CLAD) that occurs in nearly 50% of patients by the fifth post-transplant (Tx) year [17]. 

According to a recent classification, CLAD presents two major clinical phenotypes: the so-called obstructive form of bronchiolitis obliterans syndrome (BOS, nearly 70% of cases) [18], and the newly-described restrictive allograft syndrome (RAS) [19,20]. While being the most frequent one, the exact pathogenesis of BOS is still unknown, although the neutrophilic component seems to play a significant role in the maintenance of this condition. It has been speculated that BOS results from several noxious triggers which cause innate/adaptive immune reactions, leading to immune activation (mainly T helper 17, Th-17) responsible for inducing the neutrophilic inflammation characteristic of the disorder. Given the tissue-protective and anti-inflammatory properties of AAT, the interest in its influence in early and long-term complications post lung transplant has increased in recent years, also in light of the poor knowledge of complications which involve a high neutrophil recruitment, e.g., ischemia reperfusion injury and BOS [21,22,23,24,25,26,27]. In this context, BALf seems to meet the requirements for being an important tool for the study of changes in cell type and/or number and solutes that occur in the lower respiratory tract of lung transplant recipients. Identification in this fluid of potential biomarkers of BOS development/evolution would indeed offer a unique opportunity to actively intervene on the progression of the disease [28].

To obtain a global picture of the protease/antiprotease balance in the lungs of transplant recipients, BALf samples from these patients were investigated in this work. Data from electrophoretic and western blotting analysis were compared with those acquired from liquid chromatography–mass spectrometry (LC–MS). The neutrophil count, together with the determination of HNE and of Cat. G activities provided information on the functionality of AAT, its balance with HNE, and its correlation to the degree and type of inflammation of the deep graft tissue.

## 2. Materials and Methods

### 2.1. Reagents

Antibodies for detection of AAT and secondary anti-mouse antibodies were obtained from Abcam (Cambridge, UK). HNE, together with a bicinchoninic acid (BCA) protein assay kit were obtained from Thermo Scientific (Rockford, IL, USA). The standard *p*-nitroaniline (*p*-NA) and the peptide substrates used for the determination of human HNE and Cat. G activities (MeOSuc-Ala-Ala-Pro-Phe-NA and Suc-Ala-Ala-Pro-Phe-NA, respectively) were from Bachem (Bachem AG, Bubendorf, Switzerland). Unless otherwise stated, all other analytical grade reagents were purchased from Sigma-Aldrich (St. Louis, MO, USA). Double-distilled water used for the preparation of all buffers was prepared with a Millipore (Bedford, MA, USA) Milli-Q purification system.

### 2.2. Patients

The lung transplanted patients (*n* = 13) investigated in this study were enrolled at the Pneumology Unit of the IRCCS Policlinico San Matteo Foundation, Pavia, Italy. Based on their clinical features, they were classified as stable (S), potential BOS (BOS0p), BOSII (BOSII) and BOSIII (BOSIII) patients. Stable were individuals that, at >2 years post-transplantation, came up with still stable lung function, in the total absence of acute rejection or infection. Diagnosis of BOS and of its grades of severity were assessed according to published guidelines [29,30,31]. The current classification of BOS severity is based on changes in the forced expiratory volume in the first second (FEV1) and is indicated as BOS0p if FEV1 is 81–90% of the best FEV1 value obtained after transplantation; BOSI (patients not considered in this study) when FEV1 is 66–80% of the best value; BOSII when FEV1 is 51–65% of the best, and BOSIII if FEV1 is ≤50%. Individuals under investigation in this study were divided in two groups according to these characteristics. The first (Group1) contained seven subjects: six S and one BOS0p; the second (Group2) six subjects: four BOSII and two BOSIII. The immune suppression (IS) protocol applied to these patients was reported elsewhere [32]. All of them underwent surveillance and bronchoscopy at 1, 3, 6, 12, and 24 months plus on clinical need, which included the decline of lung function, and at diagnosis of chronic lung rejection. Biopsy-proven episodes of acute rejection (AR) [33] were treated with steroid boluses and, in case of AR recurrence or persistence, with a standard anti-thymoglobulin course and a modulation of the IS regimen. The surveillance protocol was reported elsewhere [34]. Patients diagnosed with BOS0p were prescribed a three-month course of chronic low-dose azithromycin. At the same time, patients underwent a gastro-esophageal reflux assessment and a maximization of anti-reflux medical treatment. In case of a further decline consistent with BOSI diagnosis, since 2003 patients have been referred to the Apheresis Unit for compassionate ECP (extracorporeal photopheresis) treatment [35]. Additionally, the cytomegalovirus surveillance protocol was detailed elsewhere [36]. Patients enrolled for this study were investigated for α1-antitripsin deficiency (AATD) at the time of listing for lung transplantation according to standard algorithm [37]. None of them resulted positive for intermediate or severe AATD. All transplanted patients were given a low-dose steroid treatment (0.05–0.1 mg/kg body weight of prednisone) as a part of the triple immunosuppressive regimen. Given that all patients were submitted to the same treatment, this was not expected to have any influence on the measurements performed on samples analyzed. All patients gave their informed consent to BALf collection.

The demographic and clinical features (including age, gender, CLAD occurrence and treatment strategies) of patients considered in this study are detailed in Table 1. 

### 2.3. BALf Collection and Processing

BALf collection was performed as previously described [28]. Briefly, the distal tip of the bronchoscope was wedged into the middle lobe or lingular bronchus and a total of 150 mL of warm sterile saline solution was instilled in five subsequent 30 mL aliquots which were sequentially retrieved by gentle aspiration. The first aliquot collected (20 mL) was used for a series of analyses which included microscopic and cultural examination of common bacteria and fungi and direct/cultural investigations for respiratory viruses. The returned fluid from the second to the fifth aliquots was pooled and further processed as BALf. Cells were recovered by centrifugation at 1500× *g* rpm for 10 min and supernatant divided in aliquots (30 mL each) which were stored at −80 °C immediately after processing, until use. 

### 2.4. AAT Measurement

AAT was measured in BALf by a rate immune nephelometric method (Immage 800 Immunochemistry System, Beckman-Coulter, Brea, CA, USA).

### 2.5. BCA Protein Assay

The exact protein concentration in each sample was determined by applying the bicinchoninic acid (BCA) assay [38] using bovine serum albumin (BSA), in the range of concentration between 5 and 25 μg/mL, to produce the calibration curves. 

### 2.6. 1D-PAGE

An aliquot of each sample (20 µg of protein) was submitted to protein precipitation with trichloroacetic acid (TCA), according to Yvon et al. [39]. After centrifugation, the pellet was reconstituted in 10 μL of 50 mM Tris–HCl pH 8.3 containing 5% 2-mercaptoethanol, 2% sodium dodecylsulphate (SDS), 0.1% bromophenol blue (BPB) and 10% glycerol. Samples were incubated at 90 °C for 10 min and then loaded on gel slabs. Electrophoresis was performed according to Laemmli [40] in 5% stacking gel and 12.5% running gel by applying a voltage of 150 V for 1 h. Gels were stained with colloidal Coomassie G-250, according to Candiano et al. [41]

### 2.7. Western Blotting

Ten micrograms of proteins were precipitated by addition of 1.22 M trichloroacetic acid (TCA) and the pellet recovered after centrifugation was submitted to sodium dodecyl sulphate - polyacrylamide gel electrophoresis (SDS-PAGE). Protein bands were transferred onto a Millipore polyvinyldivinyl fluoride (PVDF) membrane (Billerica, MA, USA) by using a trans blot turbo system (BioRad). After 1 h incubation in 5% milk diluted in phosphate buffer saline (PBS) and three washes with phosphate buffer saline containing 0.1% Tween 20 (PBST), the membrane was incubated overnight with AAT antibody (ab9400; Abcam, Cambridge, UK) at a 1:2500 dilution in 1% milk. The membrane was washed three times with PBST (10 mL), incubated with the secondary antibody, rabbit anti-mouse IgG H&L (HRP) (ab6728, Abcam), at a 1:2000 dilution in 1% milk in PBST, for 1 h at room temperature. The membrane was washed again (three times) with PBS and incubated in ECL Westar ηC Ultra (Cyanagen, Bologna, Italy) solution according to the provided protocol. The same procedure was applied for the identification of free and complexed HNE by using the anti HNE antibody (PA5-29659, Thermo Scientific) at a 1:1000 dilution. 

All immunoblots were acquired with the ImageQuant LAS 4000 analyzer (GE Healthcare Chicago, IL, USA).

### 2.8. Enzymatic Assays

The determination of HNE and Cat. G activities was carried out by applying a previously described colorimetric assay [42,43,44]. The original sample solution was exchanged into the incubation buffer by withdrawing an aliquot of each sample (400 µL), which was lyophilized and the pellet taken up in 400 µL of 50 mM Tris HCl pH 7,8 containing 500 mM NaCl. The reaction was started by the addition of 5 µL of the appropriate substrate, i.e., MeOSuc-Ala-Ala-Pro-Phe-NA for HNE and Suc-Ala-Ala-Pro-Phe-NA for CatG. The final concentrations were 2 mM and 20 mM respectively. The mixture was incubated at 37 °C for 10 min and the reaction was stopped by addition of 40 µL of 0.27 M TCA. The values of absorbance at 410 nm provided the amount of *p*-nitroaniline (*p*-NA) released by peptide hydrolysis. The limit of detection (LOD) for *p*-NA was <1 µM. One unit of enzyme activity was defined as the amount of enzyme required for the release of 1 µmol/min of *p*-NA.

### 2.9. In-Situ Digestion

Enzymatic digestion of proteins was performed as previously described [45]. Briefly, the selected bands were carefully excised from the gel, placed into eppendorf tubes, broken into small pieces and washed with 100 mM ammonium bicarbonate (AmBic) buffer pH 7.8 containing 50% acetonitrile (ACN) until complete de-staining was achieved. The gel pieces were then dehydrated by adding 200 µL of ACN until they became opaque-white color. Acetonitrile was finally removed, gel pieces were dried under vacuum for 10 min and then rehydrated by adding 75 μL of 100 mM AmBic buffer pH 7.8, containing 20 ng/μL sequencing grade trypsin (Promega, Madison, WI, USA). The digestion was performed overnight upon incubation of the mixture at 37 °C and the resultant peptides were extracted from gel matrix by a three-step sequential treatment with 50 μL of 50% ACN, 5% trifluoroacetic acid (TFA) in water and finally with 100% ACN. Each extraction involved 10 min of stirring followed by centrifugation and removal of the supernatant. All supernatants were pooled, dried and stored at −80 °C until mass spectrometric analysis. At the moment of use, the peptide mixture was solubilized in 0.1% formic acid (FA).

### 2.10. Liquid Chromatography Tandem Mass Spectrometry (LC–MS/MS)

Analyses were performed on a liquid chromatography-mass spectrometry system (Thermo Finnigan, San Jose, CA, USA) consisting of a thermostated column, a surveyor auto sampler controlled at 25 °C, a quaternary gradient surveyor MS pump equipped with a diode array (DA) detector, and a linear trap quadrupole (LTQ) mass spectrometer with electrospray ionization (ESI) ion source controlled by Xcalibur software 1.4 (Thermo Fisher Scientific, Waltham, MA, USA). Analytes were separated by reverse-phase high performance liquid chromatography (RP-HPLC) on a Jupiter (Phenomenex, Torrance, CA, USA) C18 column (150 × 2 mm, 4 µM, 90 Å particle size) using a linear gradient (2–60% solvent B in 60 min) in which solvent A consisted of 0,1% aqueous FA and solvent B of ACN containing 0.1% FA. Flow rate was 0.2 mL/min. Mass spectra were generated in positive ion mode under constant instrumental conditions: source voltage 5.0 kV, capillary voltage 46 v, sheath gas flow 40 (arbitrary units), auxiliary gas flow 10 (arbitrary units), sweep gas flow 1 (arbitrary units), capillary temperature 200 °C, tube lens voltage −105 V. MS/MS spectra, obtained by collision-induced dissociation (CID) studies in the linear ion trap, were performed with an isolation width of 3 Th *m*/*z*, the activation amplitude was 35% of ejection RF amplitude that corresponds to 1.58 V. Data processing was performed using Peaks studio 4.5 software. 

### 2.11. Workflow of the Procedure Followed in the Present Study

The workflow shown in Figure 1 summarizes the key steps of this study.

## 3. Results

### 3.1. Identification of the HNE-AAT Complex

The results of the electrophoretic runs performed on the BALf of all subjects investigated are shown in Figure 2, panel A. Lanes 1 to 7 refer to the subjects grouped in Group 1 (Stable and BOS 0p) and 8 to 13 to those in Group 2 (BOSII and BOSIII). It can be observed that two bands, at approximately 80 and 55 kDa, exceed in abundance over others. Information on these two bands was obtained by blotting the gels on a PVDF membrane that was later incubated with an anti-AAT antibody. The results are shown in panel B of Figure 2. Based on the presence/absence of the immune-reactive band at 80 kDa, individuals belonging to Group 1 could be divided in two sub-classes: those who did not exhibit this band (lanes 1 to 3) and those who showed it (lanes 4 to 7). Conversely, the image of the PVDF membrane obtained from patients belonging to Group 2 (lanes 8 to 13), mirrored that of the starting gel. The 80 kDa band was evident, although at different intensity, in all profiles, regardless of whether the BALf belonged to BOS II or BOS III patients. Based on the well-known molecular mass of AAT (Mr 52,0 kDa), the band at approximately 55 kDa was attributed to this protein. Likewise, on the basis of the sum of the theoretical molecular weights of AAT and HNE, the band at 80 kDa was tentatively attributed to the complex between these two proteins. While experimental evidence was still lacking, the reactivity of the material under this band against the anti-AAT antibody was circumstantial evidence that it contained AAT. 

The presence of a band at approximately 80 kDa in SDS-PAGE had been previously observed in vitro also by other authors who speculated that it could correspond to the HNE-AAT complex [46,47]. To definitively prove the existence of this inhibitory complex in BALf samples that showed this band, it was carefully excised and the protein submitted to the procedure detailed in the experimental section. Peptides generated from tryptic digestion were separated by LC–MS/MS and fragmentation data searched against the Swiss-Prot database [48,49]. Data relative to the proteins present under the 80 kDa band (shown in Table 2) allowed us to unambiguously identify both HNE and AAT. The same procedure was applied to the band at approximately 55 kDa to confirm that it contained AAT. The data of Table 2 show that this band contained AAT contaminated by proteins with similar molecular weight which co-migrated with the former due to the limits of resolution of 1D electrophoresis. Additional information concerning the primary sequence of all peptides identified for each protein analyzed is included in Appendix A.

### 3.2. Determination of AAT Amount and of Protease Activities 

To answer the question of whether HNE had fully complexed AAT or was still partially present in BALf as a free, active enzyme, the activity of HNE was determined in all samples. The results summarized in Table 3 pointed out that significant levels of HNE activity were detectable only in BALf samples presenting the 80 kDa band. Conversely, the amount of AAT did not differ significantly among samples. 

Being that the Cat. G activity levels are lower than the LOD of the procedure, they could be estimated only in two samples (samples 8 and 9).

As shown in Figure 3, a reasonable correlation was observed between the specific activity of HNE and the number of neutrophils determined in all samples.

### 3.3. Use of Western Blotting for HNE Detection

The presence/absence of HNE in the BALfs of subjects investigated was checked by running the samples on SDS-PAGE and blotting the bands on a PVDF membrane that was incubated with anti-HNE antibody (Figure 4). The absence of immune reactive bands in BALfs lacking the 80 kDa band (lanes 1 to 3) unambiguously demonstrates that these samples do not contain HNE. Conversely, the specimens that formed the complex showed the presence of two bands (lanes 4 to 13). Based on their migration on the gel, these bands were assigned to free and complexed HNE. Following the LC–MS procedure described above, HNE was identified in both bands (data not shown).

### 3.4. Inhibitory Capacity of AAT

The inflammation-mediated cell oxidative stress was previously observed to inactivate in vitro AAT through a sort of oxidative cascade [46,47]. This would prevent (partially or totally) its capacity to inhibit HNE. The efficiency of AAT towards HNE was checked in BALfs from subjects who did not show the HNE-AAT complex by adding exogenous HNE (2 µL; 0.2 mg/mL) to samples. After 15 min of incubation at 37 °C, an aliquot was submitted to SDS-PAGE and gels blotted on a PVDF membrane which was incubated with the anti-AAT antibody. Figure 5, panel A, illustrates the results relative to these samples before (lanes 1; 2 and 3) and after incubation (lanes 1 *; 2 * and 3 *). The formation of the 80 kDa band clearly indicated that the BALf sample was able to capture the “excess” of HNE thus suggesting that an amount of functional AAT was still present in these samples. 

When the same amount of exogenous HNE was added to samples which presented the complex, the 80 kDa band was seen to increase considerably. Figure 5, panel B, shows the results relative to a sample chosen at random among all available (sample #12). The results relative to other samples are shown in Appendix A. These data further support the hypothesis that AAT was functional in these samples.

## 4. Discussion

Previous studies had already reported that BOS progression is associated with a protease/antiprotease imbalance [16]. To get insights into this mechanism of the control of inflammatory lung diseases, possible variations in the protease/antiprotease balance of BALf from two different cohorts were evaluated. Lung transplant recipients, both stable and developing BOS after lung transplantation, have been investigated. Since the mere assay of HNE and AAT involved in the process is not able to provide information on the level of activity of these proteins, we explored their capacity to interact. Given the role of AAT and neutrophil elastase being its main physiological target, the formation of a complex between them must be interpreted as the result of the protective effect of this inhibitor against the proteolytic action of the protease. The existence of this complex in the BALf samples of cystic fibrosis patients had previously been demonstrated by a double-ligand ELISA [50]. The finding that a few samples of stable post-transplant subjects lacked this inhibitory complex was unexpected and opened the door to a series of questions. In fact, although the Western Blot analysis clearly indicated that all samples contained AAT, it did not provide information about the ability of this inhibitor to bind HNE in whole or in part. Thus, whether the lack of such complex in the BALf of the mentioned subjects should be ascribed to their pathological state, i.e., the low level of pulmonary inflammation, remained a speculation. In fact, while protease activity was undetectable in patients with a low level of lung inflammation, the formation of the complex could also have been prevented by possible inactivation processes at the expense of AAT occurring in the lungs of these individuals. This latter hypothesis was denied by the addition to the above samples of exogenous HNE, which lead to the formation of the complex. Instead, our findings supported the assumption that AAT was, at least in part, functional.

This result arouses a major question. On the assumption that any pathological condition leads to a number of pulmonary physiological changes, could this complex be considered a biomarker of lung status? It seems logical to argue that free HNE would be a marker of the degree of injury more relevant than the complex. In fact, if the rate of formation of AAT–HNE complexes is limited by the amount of available AAT, then it would be easier to measure NE in excess in the lung. The fact that free elastase in inflamed tissues can only be detected if it is present in excess over AAT activity or if this anti-elastolytic function has been somehow modified, was previously observed by other authors [51,52]. This is not the case of the patients under investigation. In fact, given that both the initial lack of the complex and its formation by addition of exogenous HNE cannot reflect with certainty an excess of the protease, our results point to a different conclusion. The unifying view that apparently emerges from these data is that not necessarily all recipients classified as “stable” based on their clinical/functional status, display the same conditions at a molecular level. The neutrophil count and detection of active HNE in tissues and fluids recovered from inflammatory sites might represent a critical step in tissue pathogenesis. In fact, significantly lower neutrophil levels in BALf from healthy subjects compared to BOS patients and the tendency of both HNE activity and the concentration of HNE-AAT complex to increase in these latter, had been previously observed [16]. Neutrophils produce antioxidants and are indispensable in forming the first line of defense during infection. In addition to the well-known anti-protease traits, evidence has been increasing in the last years that AAT has tissue-protective, anti-inflammatory and immune-regulating properties that can affect most inflammatory cells [22]. As a consequence, the concern that AAT has a great influence in early and long-term outcomes post lung transplantation has gained much interest [53]. Despite this interest, knowledge about the complications of high neutrophil recruitment is still poor and bronchoscopy with bronchoalveolar lavage seems to be the best diagnostic approach to investigate the local alterations at bronchial and alveolar levels. As shown in Table 3, while the number of neutrophils was very low in the BALf of the three stable individuals who did not present free elastase activity or complex formation, these numbers and the HNE activity were increasing along with the severity of the disease. In this context, it is no wonder that our attention has also been drawn to the investigation of AAT functionality. It has been recently reported that macrophages or neutrophils have the potential ability to alter its activity in a manner that would allow neutrophil elastase to accumulate [54]. However, while several studies have assumed that oxidation of AAT promotes detection of free HNE activity in vivo [46,47], very poor experimental evidence still supports this contention. The reason is that almost all studies of AAT function after its oxidation in vitro or in vivo have been limited to the determination of its ability to inhibit exogenous porcine pancreatic elastase [54]. Obviously, our data do not allow us to exclude the fact that the lack of formation of the complex in some of our BALf samples was due to AAT modifications that made it unable to completely inhibit neutrophil elastase activity. However, the addition of exogenous HNE to samples not showing the 80 kDa band and the consequent formation of this complex seemed to show the functionality of AAT present in these samples. The great increase observed in the 80 kDa band by addition of exogenous HNE was a further evidence that this hypothesis was most likely correct. 

## 5. Limitations of the Study

The sample size of individuals investigated represents a limitation of the present study. We would like to note that this research was conceived as a feasibility study whose aim was of acquiring a global vision of the protease/antiprotease balance in patients affected by BOS at different levels of severity. In an effort to work with well-characterized cohorts, great attention was paid to excluding sources of excessive variability and this choice necessarily lessened the sample size. Thus, the distribution of single versus double lung transplant procedures was similar, recipients affected by AATD were excluded, as were similar types of transplant indications, analogous IS regimen, NO azithromycin treatment at time of sampling, and appreciable length of post-transplant FU for enrolled stable recipients. The observed variability in sampling time is related to the evolution of BOS, whose grading reflects disease severity. While in some cases the evolution towards more severe disease is fast, in other cases patients evolve to BOSIII grade more slowly. Patients numbers 12 and 13 had been sampled after several months of disease course. The severity of disease however was not necessarily associated with a higher degree of inflammation. 

That the investigation was limited to subjects with stable pulmonary function and patients affected by BOS II and BOS III can be considered another limitation of this work. However, the decision of not considering the BOSI patients at this stage of the work reflects the abovementioned need to avoid potential sources of variability. In fact, the possible reversibility of pulmonary alterations, common to this stage of BOS, could represent a confounding factor that may have affected the results. Finally, the demographic data shown in Table 1 might highlight the heterogeneity of the lung transplant recipients considered in this study. We consider this hypothetical heterogeneity not relevant for the purposes of our work. In fact, we focused our investigation only on the assessment of the AAT/HNE balance at graft level in patients submitted to double lung transplantation that were influenced only by graft acceptance/rejection status.

Of course, we are aware that a high-quality set of samples does not necessarily eliminate the risk of relying on poor evidence of data and that, to obtain more concrete answers about a condition that is largely still shrouded in mystery, larger-scale studies will be essential. 

These results are preliminary and must, obviously, be validated on a larger cohort of patients that should involve individuals with different levels of BOS severity and co-morbidities associated with the presence of chronic rejection that could affect the protease/antiprotease balance. 

## 6. Conclusions

This pilot work aimed to obtain a global picture of the protease/antiprotease balance in the BALf of two different cohorts of lung transplant recipients. The results of our study allowed us to demonstrate, for the first time, that not all BALf samples contain the complex between HNE and its specific inhibitor AAT. The lack of this complex in some subjects seems to be correlated with the low level of pulmonary inflammation, as suggested by the count of neutrophils and the determination of HNE/AAT activities in these subjects. The confirmation of these data on a larger cohort of individuals would allow us to consider the AAT-HNE complex a marker of pulmonary inflammation. This could indeed represent a new frontier in the field of biomarker discovery of lung inflammation in subjects with HNE/AAT unbalance. The identification of markers that could predict the development and progression of BOS is crucial for preventing the irreversible phase of the disorder with the final goal of improving the survival of lung transplant recipients.

## Figures and Tables

**Figure 1 high-throughput-08-00005-f001:**
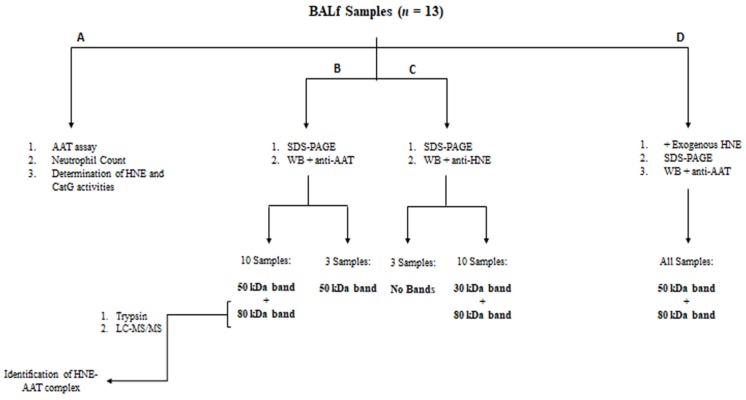
Workflow of the experimental procedure.

**Figure 2 high-throughput-08-00005-f002:**
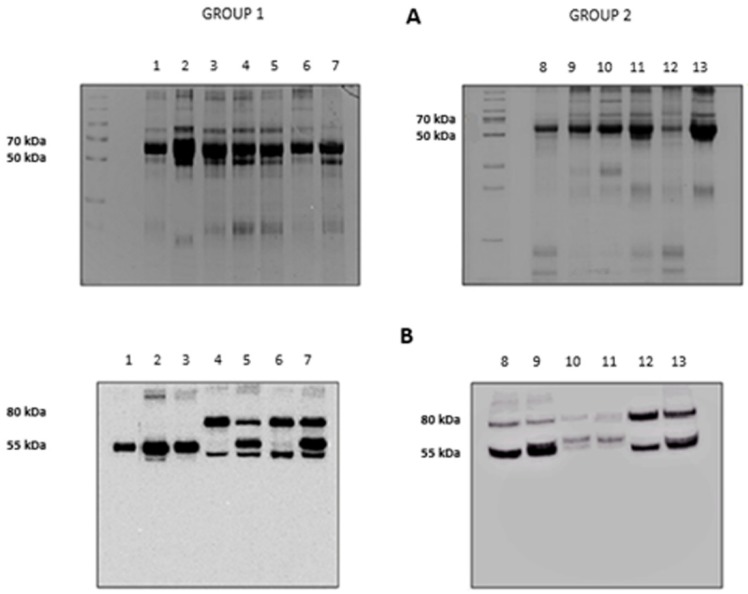
Panel A: 12.5% SDS-PAGE showing the protein profile of all the bronchoalveolar lavage fluid (BALf) samples considered. Group 1: lanes 1–7, Group 2: lanes 8–13. Panel B: Western blotting with anti-AAT antibody of the same samples as in panel A.

**Figure 3 high-throughput-08-00005-f003:**
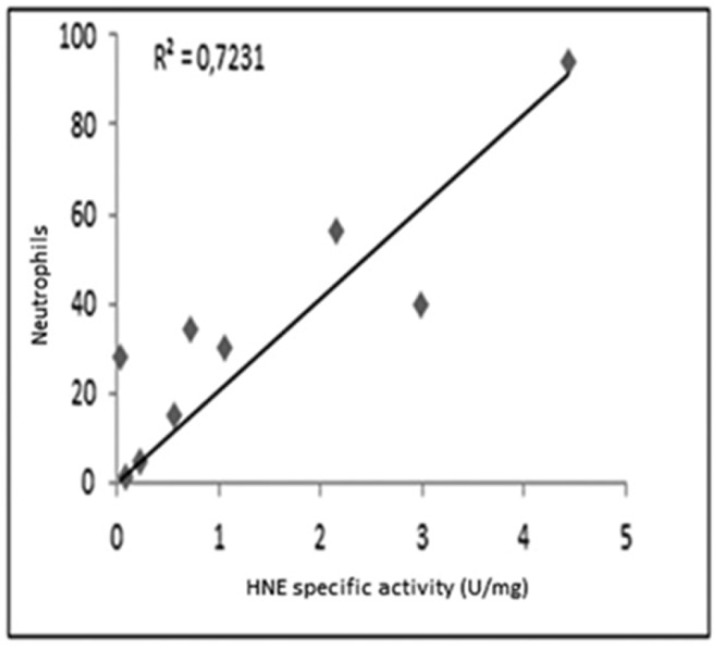
Correlation between the specific activity of HNE and the count of neutrophils in samples analyzed.

**Figure 4 high-throughput-08-00005-f004:**
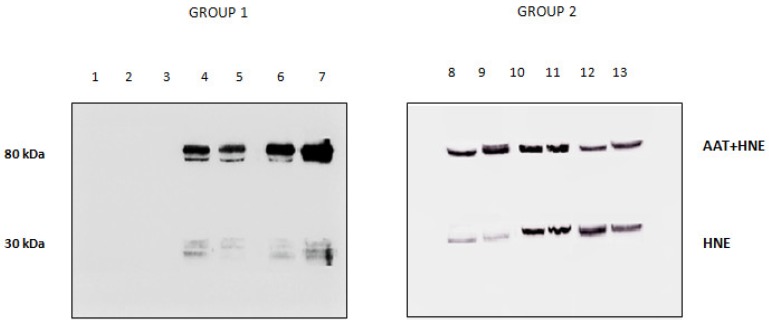
Western blotting profile obtained upon incubation of all BALfs with the anti-HNE antibody. Lanes 1–3: samples not showing the 80 kDa band when incubated with anti-AAT. Lanes 4–13: sample showing the 80 kDa band.

**Figure 5 high-throughput-08-00005-f005:**
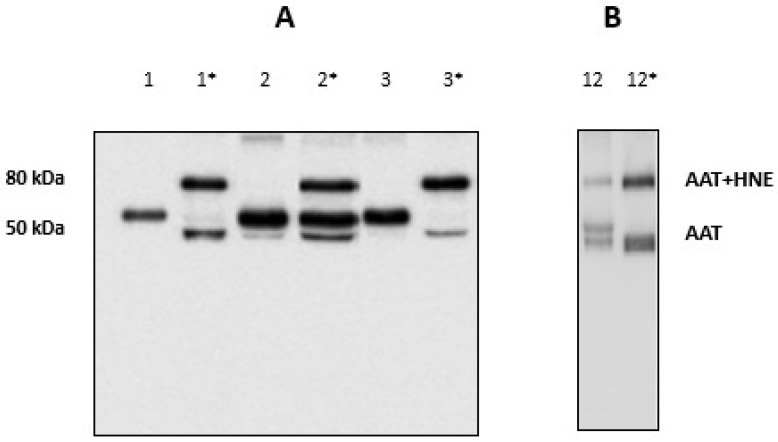
(**A**) Western blotting of BALfs from the three patients without the 80 kDa band, before (1; 2 and 3) and after incubation with exogenous HNE (1 *; 2 * and 3 *). (**B**) Western blotting of BALf from a patient with the 80 kDa band, before (12) and after incubation with exogenous HNE (12 *).

**Table 1 high-throughput-08-00005-t001:** Demographic data of individuals considered in this study.

Sample #	Age	Sex	CLAD	Months from CLAD	Azithromycin	ECP	Immunosuppressive Therapy
1	65	M	Stable	n.d. *	No	No	CsA, AZA
2	51	M	Stable	n.d. *	No	No	TAC, MMF
3	64	M	Stable	n.d. *	No	No	TAC, MMF
4	56	M	Stable	n.d. *	No	No	CsA, AZA
5	34	M	Stable	n.d. *	No	No	CsA, AZA
6	57	M	Stable	n.d. *	No	No	CsA, AZA
7	42	M	BOS0p	13.57	Yes	No	TAC, MMF
8	62	M	BOS2	1.00	No	No	CsA
9	28	M	BOS2	2.57	Yes	No	TAC, MMF
10	62	M	BOS2	1.00	No	No	TAC, MMF
11	63	F	BOS2	1.00	Yes	No	TAC, AZA
12	26	M	BOS3	23.43	Yes	Yes	TAC
13	53	F	BOS3	38.87	No	Yes	CsA, RAD

* not determined. CsA: Cyclosporin A; TAC: Tacrolimus; RAD: Everolimus; MMF: Mycofenolate Mofetil; AZA: Azathioprine; CLAD: chronic lung allograft dysfunction. None of the samples were obtained during Azithromycin treatment.

**Table 2 high-throughput-08-00005-t002:** Proteins identified by liquid chromatography–mass spectrometry (LC–MS/MS) under bands at 50 and 80 kDa.

Molecular Weight	Accession	Mass (kDa)	Score (%)	Coverage (%)	Description
**80 kDa**	sp|P01009|A1AT_HUMAN	46,737	67	9.33%	α1-antitrypsin OS = *Homo sapiens* GN = SERPINA1 PE = 1 SV = 3
	sp|P08246|ELNE_HUMAN	28,518	62	3.37%	Neutrophil elastase OS = *Homo sapiens* GN = ELANE PE = 1 SV = 1
**55 kDa**	sp|P02768|ALBU_HUMAN	69,367	98	11.82%	Serum albumin OS = *Homo sapiens* GN = ALB PE = 1 SV = 2
	sp|P01009|A1AT_HUMAN	46,737	85	6.70%	α1-antitrypsin OS = *Homo sapiens* GN = SERPINA1 PE = 1 SV = 3
	sp|P01859|IGHG2_HUMAN	35,901	60	4.29%	Ig γ-2 chain C region OS = *Homo sapiens* GN = IGHG2 PE = 1 SV = 2

**Table 3 high-throughput-08-00005-t003:** Biochemical features of all samples analyzed.

Sample #	Clinical Classification of the Disorder	AAT Assay (mg/dL)	Neutrophil Count	Elastase Specific Activity (mU/mg)	Cathepsin G Specific Activity (mU/mg)	Presence of the 80 kDa Complex
1	Stable	0.57 *	1	n.d. **	n.d. **	NO
2	Stable	1.00	1	0.08	n.d.	NO
3	Stable	0.32	1	n.d.	n.d.	NO
4	Stable	0.79	34	0.725	n.d.	YES
5	Stable	0.37	30	1.06	0.14	YES
6	Stable	0.40	1	n.d.	n.d.	YES
7	BOS 0p	0.26	28	0.29	n.d.	YES
8	BOS II	2.50	94	4.44	0.57	YES
9	BOS II	0.56	56	2.16	0.254	YES
10	BOS II	0.12	40	2.98	n.d.	YES
11	BOS II	0.12	5	2.23	n.d.	YES
12	BOS III	0.13	10	n.d.	n.d.	YES
13	BOSIII	0.13	15	1.56	n.d.	YES

* Values reported are the mean of three independent determinations. Standard deviation was within 5%. ** n.d. = not detectable.

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
