# Peer review of "A Pilot Study to Investigate the Balance between Proteases and α1-Antitrypsin in Bronchoalveolar Lavage Fluid of Lung Transplant Recipients"

_2571-5135, 2019, doi:10.3390/ht8010005_

Round 1

Reviewer 1 Report

In this pilot study, Cagnone and coworkers investigated the balance between elastase and antitrypsin in BAL of a small cohort of lung transplant recipients with BOS. Despite being a retrospective analysis, and limited patient numbers, their findings are novel and interesting to the field. 

Major comments: 

- Table 1: there is a wide variability in time of BAL sampling since CLAD (column 5), with samples in BOS 2 taken mucht earlier after BOS onset compared to tho those in BOS 3: please explain this difference in timing of sampling. Did this time of sampling correlate with any of the measured BAL characteristics (neutrophil count, Elastase activity,...)? 

- Table 1: please clarify if (Methyl-Prednisolone was part of the immunosuppressive therapy in the included patients (or only TAC/CsA, AZA/MMF).

- Table 3: there are 2 stable patients with high neutrophil count: did these patients later (after sampling) develop to CLAD/BOS? If not, where can this high neutrophilia be attributed to (in abcense of concurrent infection, which was presumably exluded)?

- Conclusion, P14: It is incorrect to state that 'based on the presence/abcense of this complex, they could be discriminated in two sub-groups', this is definitely not supported by the results.

The only thing that is shown is a relation (association) of AAT-HNE complex with neutrophilic inflammation, which should be the only conclusion of these findings.

Minor comments: 

- P2, line 58: Padjak and Coll. To should be: Padjak and coworkers to 

- P2, line 63: The aim 

- P 3, line 96: secondary anti-mouse antibodies

- P3, line 111: 81-90% of the average of the two best FEV1 values 

- P13, line 450: please rephrase 'is greatly growing up' by 'has gained much interest'

- P13, line 471: graft

- P13, line 483: please replace 'full-blown' by 'severe'

Author Response

Point-by-point answers to the reviewers’ concerns:

Reviewer 1

1.       Table 1: there is a wide variability in time of BAL sampling since CLAD (column 5), with samples in BOS 2 taken mucht earlier after BOS onset compared to tho those in BOS 3: please explain this difference in timing of sampling. Did this time of sampling correlate with any of the measured BAL characteristics (neutrophil count, Elastase activity,...)?

The variability in sampling time is related to the evolution of BOS whose grading reflects disease severity . While in some cases evolution towards more severe disease is fast, in other cases patients evolve to BOSIII  grade more slowly. Patients n° 12 and 13 had been sampled after several months of disease course. The severity of disease however is not necessarily associated to a higher degree of inflammation.

This new sentence was introduced in the text.

2.       Table 1: please clarify if (Methyl-Prednisolone was part of the immunosuppressive therapy in the included patients (or only TAC/CsA, AZA/MMF).

All transplanted patients are given a low dose steroid treatment (0.05-0.1 mg/kg body weight) as part of the triple immunosuppressive regimen. Since the treatment is the same for all patients we  do  not expect any influence on our measures  in different disease groups.

             This new sentence was introduced in the text.

3.       Table 3: there are 2 stable patients with high neutrophil count: did these patients later (after sampling) develop to CLAD/BOS?

Yes, both patients developed subsequently BOS  (n°  5 8 months after sampling and n° 6  10 months). At time of sampling were still  functionally stable.

If not, where can this high neutrophilia be attributed to (in absence of concurrent infection, which was presumably excluded)?

Neutrophilia was not due to infection since the presence of positive BAL cultures represented an exclusion criterion in this study. Authors believe that neutrophilia was related to BOS pathogenesis and predated BOS occurrence.

4.       Conclusion, P14: It is incorrect to state that 'based on the presence/absence of this complex, they could be discriminated in two sub-groups', this is definitely not supported by the results.

            This paragraph was totally modified

5.       Minor Comment:

P2, line 58: Padjak and Coll. To should be: Padjak and coworkers to

P2, line 63: The aim

P3, line 96: secondary anti-mouse antibodies

P3, line 111: 81-90% of the average of the two best FEV1 values

P13, line 450: please rephrase 'is greatly growing up' by 'has gained much interest'

P13, line 471: graft

P13, line 483: please replace 'full-blown' by 'severe'

All changes suggested by the reviewer have been introduced in the revised version.

Reviewer 2 Report

In their manuscript the authors describe the AAT and HNE balance in lung transplant patients' BALf and they try to correlate this with BOS which is a major clinical phenotype of CLAD. The identification of a biomarker for patients at risk to develop BOS would be a big improvement for the care of lung transplant patients.

The introduction provides a lot of background information, however, it should be presented in a more expedient way. Avoid to give explanations in brackets since it interrupts the reading flow.

Page 1, line 52: exchange "substrate" by "inhibitor"

Page 1, line 60: Mistake in the reference list: citation 13 is not Pajdak and Coll. Citation 12 is missing.

It should be pointed out that many researchers examined AAT and HNE activities in BAL fluid of patients with different lung disorders and in which conditions they found a correlation of HNE and/or HNE/AAT complexes with the severity of the disease.

For the understanding of the reviewer it is exaggerated to call standard biochemical techniques like SDS PAGE, Western blot and photometrical enzymes assays in combination with cell counts and mass spectrometry "multifaceted". The only specialities in the approach are the analysis of HNE-AAT complexes by MS in excised gel pieces and the addition of HNE to BALf in order to check for the anti-elastase activity of the endogenous AAT.

Materials and Methods chapter gives a lot of details which are unnecessary for standard methods (e.g. page 4, line 164 (10 ml), however important informations (e.g. the concentration of Tween 20 (0.1 %?) and the dilution of HNE antibody) is missing. Thus, this chapter could be much more concise and precise.

For the enzymatic assays the reviewer asks for the advantage of the lyophilization of BAL. In addition, the authors shall give evidence that lyophilization doesn't influence HNE or Cat G activities and that addition of TCA didn't interfere with nitroaniline absorbance.

Page 6: The Results chapter starts with demographic data and workflow. Demographic data table should be moved into Materials and Methods chapter.

Page 7: Workflow schematic is expendable. 

Results should be presented much more concise. The information is often repetitive (see also Methods or Figure Legends). Avoid filler words (like in fact, well-known, circumstantial on page 7). Avoid circumstantial expressions (To get insight into the nature of these bands..., or However, the fact that the material under this band...). Please explain why you blot onto anti-AAT loaded PVDF membrane. Normally, the membrane is incubated with the primary antibody after blotting and blocking.

Page 8: Figure 2 misses molecular weights. Please also mark 55 and 80 kDa bands. Legend misses header. Explain what means group 1 and group 2 in the figure legend. Avoid "definitively prove". Avoid "real" in context with "real BALf samples". What do you mean with real? Be more precise! 

Why did you analyze the 55kDa band? What do you expect? Do you expect another 55kDa protein different from AAT that - by chance - cross-reacts with anti-AAT antibodies? Please explain.

Page 9: Instead of activities, anti HNE blot would be more straight forward. The authors should show Western blot of BALf from all patients. The appearance of un-complexed HNE migrating at about 30 kDa would directly point onto elastase activities in the samples. Then you can show elastase and Cat G activities and AAT concentrations and examine for imbalance of proteases and AAT.

What is the limit of detection for both assays?

Page 9, line 328: Compare the statement about correlation with the level of inflammation with the literature. Explain why you speculate about the correlation if your figure clearly shows it. For your samples you see a correlation between number and activity with correlation factor r2=0.72.

Page 11: Avoid "For simplicity". Show HNE blot for all samples. Show also inhibitory capacity of AAT for all samples. 

The Discussion must be improved. It should be concise, focus on the main findings and put these in the context of the literature. Please avoid general statements like "imbalance is very complex for the extreme complexity of this disorder". 

Page 12, line 418: The lack of the complex is, indeed, not surprising. It fits very well to the low neutrophil count and to the low or missing elastase activity.

line 427: From your data you cannot assume that AAT is "fully" functional since you didn't give access of HNE and you didn't measure inhibitor kinetics with different amounts of elastase. 

The crucial question is if the anti-elastase activity of AAT varies from sample to sample and if this correlates with severity of the disease. Therefore the authors should show HNE blot and inhibitory capacity of AAT for all samples.

Page 13: Limitations of the study are written conversational (see line 473: A few words about... line 475: We would like to note...). Scientific writing must be concise and evidence-based. 

Page 14: Conclusion resembles the rest of the paper in writing style and content. 

To sum up the paper is not acceptable for publication in the present form. Important data is missing, writing style is not scientific, language must be improved. If the authors are able to present the missing data the paper must be completely reworked. The authors should also think about to write a short communication instead of a full research paper.

Author Response

Point-by-point answers to the reviewers’ concerns:

Reviewer 2

-The introduction provides a lot of background information, however, it should be presented in a more expedient way. Avoid to give explanations in brackets since it interrupts the reading flow.

Introduction was largely modified and, as suggested, all brackets containing explanations have been cancelled to make reading more fluent.

-Page 1, line 52: exchange "substrate" by "inhibitor"

We agree. The word substrate was changed with “inhibitor”.

-Page 1, line 60: Mistake in the reference list: citation 13 is not Pajdak and Coll. Citation 12 is missing.

We apologize for the mistake. The citations, as well as the list of references have been  corrected.

-It should be pointed out that many researchers examined AAT and HNE activities in BAL fluid of patients with different lung disorders and in which conditions they found a correlation of HNE and/or HNE/AAT complexes with the severity of the disease.

New sentences ( and new references) have been added to the text to mention the studies performed on BALf of patients with different lung disorders dealing with AAT and HNE activities and their correlation with the disease severity.

-For the understanding of the reviewer it is exaggerated to call standard biochemical techniques like SDS PAGE, Western blot and photometrical enzymes assays in combination with cell counts and mass spectrometry "multifaceted". The only specialities in the approach are the analysis of HNE-AAT complexes by MS in excised gel pieces and the addition of HNE to BALf in order to check for the anti-elastase activity of the endogenous AAT.

We are perfectly aware that those applied in our study were standard biochemical techniques. The word “multifaceted” was used to underline that it was the contribution of electrophoretic, chromatographic  and spectrophotometric data that allowed to  clarify the presence and the level of functionality of HNE/AAT in our samples. However, to avoid misunderstandings, this word was cancelled and the paragraph was completely re-written.

-Materials and Methods chapter gives a lot of details which are unnecessary for standard methods (e.g. page 4, line 164 (10 ml), however important informations (e.g. the concentration of Tween 20 (0.1 %?) and the dilution of HNE antibody) is missing. Thus, this chapter could be much more concise and precise.

This section was simplified and the missing information added.

- For the enzymatic assays the reviewer asks for the advantage of the lyophilization of BAL. In addition, the authors shall give evidence that lyophilization doesn't influence HNE or Cat G activities and that addition of TCA didn't interfere with nitroaniline absorbance.

That  described is a standard procedure extensively used for the detection of proteases.  Comprehensive details about experimental conditions and potential pitfalls may be found in the  literature. We apologize for not having indicated the appropriate references in this paragraph. They have been added in the revised version together with the information concerning the LOD.

-Page 6: The Results chapter starts with demographic data and workflow. Demographic data table should be moved into Materials and Methods chapter.

As suggested, the demographic Table was moved to the Materials and Methods section, at the end of the paragraph describing patients enrollment and their medical conditions.

-Page 7: Workflow schematic is expendable. 

Sorry, we would like to maintain this scheme. Our fear is that the reader may get  confused by the great number of SDS-PAGEs and western blotting that, in all cases, refer to the same 55 and 80 kDa bands. In our opinion the workflow helps the reader to better understand the rationale of the study.

-Results should be presented much more concise. The information is often repetitive (see also Methods or Figure Legends). Avoid filler words (like in fact, well-known, circumstantial on page 7). Avoid circumstantial expressions (To get insight into the nature of these bands..., or However, the fact that the material under this band...). Please explain why you blot onto anti-AAT loaded PVDF membrane. Normally, the membrane is incubated with the primary antibody after blotting and blocking.

Results have been re-written, circumstantial expressions deleted and mistakes corrected.

- Page 8: Figure 2 misses molecular weights. Please also mark 55 and 80 kDa bands. Legend misses header. Explain what means group 1 and group 2 in the figure legend. Avoid "definitively prove". Avoid "real" in context with "real BALf samples". What do you mean with real? Be more precise! 

The figure was completed with the indication of the molecular weights. All explanations requested have been included.

- Why did you analyze the 55kDa band? What do you expect? Do you expect another 55kDa protein different from AAT that - by chance - cross-reacts with anti-AAT antibodies? Please explain.

Yes, the 55 kDa band was analyzed to confirm that it was AAT and to exclude that another protein with the same molecular weight could cross-react with the anti AAT antibody. To make the text clearer, this explanation  was added.

- Page 9: Instead of activities, anti HNE blot would be more straight forward. The authors should show Western blot of BALf from all patients. The appearance of un-complexed HNE migrating at about 30 kDa would directly point onto elastase activities in the samples. Then you can show elastase and Cat G activities and AAT concentrations and examine for imbalance of proteases and AAT.

In addition to the table containing the activity values, as suggested, the western blots of BALf from all patients have been shown in  figure 4.

- What is the limit of detection for both assays?

The limit of detection was indicated in the text.

- Page 9, line 328: Compare the statement about correlation with the level of inflammation with the literature. Explain why you speculate about the correlation if your figure clearly shows it. For your samples you see a correlation between number and activity with correlation factor r2=0.72.

Yes, indeed our experimental data show a correlation between the number of neutrophils and the HNE activity. The use of the word “speculate” was not correct. The text was changed to explain better this concept.

- Page 11: Avoid "For simplicity". Show HNE blot for all samples. Show also inhibitory capacity of AAT for all samples. 

As suggested, the figure was changed to show HNE blot. In our opinion this figure is redundant however to show our results, it has been added as Figure 1S of supplemental Material.

-The Discussion must be improved. It should be concise, focus on the main findings and put these in the context of the literature. Please avoid general statements like "imbalance is very complex for the extreme complexity of this disorder". 

Discussion was shortened and largely changed  to focus it on the main findings of the work.

-Page 12, line 418: The lack of the complex is, indeed, not surprising. It fits very well to the low neutrophil count and to the low or missing elastase activity.

We agree.

-line 427: From your data you cannot assume that AAT is "fully" functional since you didn't give access of HNE and you didn't measure inhibitor kinetics with different amounts of elastase. The crucial question is if the anti-elastase activity of AAT varies from sample to sample and if this correlates with severity of the disease. Therefore the authors should show HNE blot and inhibitory capacity of AAT for all samples

We agree. The text was modified. As indicated above, HNE blot and inhibitory capacity of AAT for all samples have been shown.

- Page 13: Limitations of the study are written conversational (see line 473: A few words about... line 475: We would like to note...). Scientific writing must be concise and evidence-based. 

This paragraph was re-written.

-Page 14: Conclusion resembles the rest of the paper in writing style and content. 

Also this paragraph was modified .

Round 2

Reviewer 2 Report

The authors improved their manuscript very much in many points.

Only a few points for corrections remain:

1. Abstract: they missed to remove real in real BALf. Last sentence should give the direction of speculation: presence OR absence?

2. Workflow chart is NOT a result, so it should NOT be presented in Results section. If the authors still think that the reader cannot understand the study without that chart it can maybe presented in the Methods section. But don't underestimate readers.

3. Page 7, line2: To make it more clear please indicate which patients are in which group either in the text, or in the figure, or in the figure legend.

4. Firgure 2 A: Both gel ran very differently. Thus indicate molecular weight of weights markers also in Group 2 gel.

5. Page 10: the last sentence doesn't make sense.

6. Page 11, line 372: Statement of the authors is wrong! It is NOT the first evidence for HNE/AAT complex in human BAL fluid.  

>See the following paper: 

Am Rev Respir Dis. 1991 Sep;144(3 Pt 1):580-5.

Human neutrophil elastase and elastase/alpha 1-antiprotease complex in cystic fibrosis. Comparison with interstitial lung disease and evaluation of the effect of intravenously administered antibiotic therapy.

Meyer KC1Lewandoski JRZimmerman JJNunley DCalhoun WJDopico GA.

7.Point out, what is really new in your study!

8. Please explain how did you exclude patients with acute infections from the study group? Infections induce secretion of AAT and other positive acute-phase proteins from the liver.

9 Please also check language again.

Author Response

Point-by-point answers to the reviewers’ concerns:

Reviewer 2

1. Abstract: they missed to remove real in real BALf. Last sentence should give the direction of speculation: presence OR absence?

We agree, Sorry. As suggested, the text was modified to give the direction.

2. Workflow chart is NOT a result, so it should NOT be presented in Results section. If the authors still think that the reader cannot understand the study without that chart it can maybe presented in the Methods section. But don't underestimate readers.

OK, the workflow was moved to the Material and Methods section

3. Page 7, line2: To make it more clear please indicate which patients are in which group either in the text, or in the figure, or in the figure legend.

This indication was added to the text 

4. Figure 2 A: Both gel ran very differently. Thus indicate molecular weight of weights markers also in Group 2 gel.

As suggested the figure was completed

5. Page 10: the last sentence doesn't make sense.

Sorry, a portion of the sentence was lost. It has been  re-written.

6. Page 11, line 372: Statement of the authors is wrong! It is NOT the first evidence for HNE/AAT complex in human BAL fluid.

The text was modified according to the reviewer suggestions and a new reference added.

7.Point out, what is really new in your study!

The conclusions contain a new sentence pointing out the new data of our work.

8. Please explain how did you exclude patients with acute infections from the study group? Infections induce secretion of AAT and other positive acute-phase proteins from the liver.

Since we were aware that infections induce a misbalance in alpha1 antitrypsin/elastase ratio, we excluded all samples with positive microbiological results and/or from patients with sign and symptoms of respiratory infection.

9 Please also check language again.

The text was edited by a native English speaking